# Examining Stakeholder Perspectives: Process, Performance and Progress of the Age-Friendly Taiwan Program

**DOI:** 10.3390/ijerph16040608

**Published:** 2019-02-19

**Authors:** Li-Ju Lin, Yu-Chang Hsu, Andrew E. Scharlach, Hsien-Wen Kuo

**Affiliations:** 1Health Promotion Administration, Ministry of Health and Welfare, Taipei 10341, Taiwan; lilian@hpa.gov.tw (L.-J.L.); plyu0703@hpa.gov.tw (Y.-C.H.); 2School of Social Welfare, UC Berkeley, Berkeley, CA 94720-7400, USA; scharlach@berkeley.edu; 3Alliance of Healthy Cities in Taiwan, Taipei 11221, Taiwan; 4Institute of Environmental and Occupational Health Sciences, National Yang Ming University, Taipei 11221, Taiwan; 5School of Public Health, National Defense Medical Center, Taipei 1141, Taiwan

**Keywords:** perception, age-friendly, facilitators, experts

## Abstract

Since Taiwan’s age-friendly city (AFC) program was launched in 2012, the central government has provided various resources to the country’s 22 local authorities, including budgetary support, policy advocacy, and consultation from a team of experts. This study examines stakeholder perspectives on the process, performance, and outcome of the AFC program. A 53-item questionnaire was developed based on the World Health Organization (WHO) guideline, including mechanisms and processes (20 items), outcome evaluations (23 items), and resource integration (10 items). There was a “great difference” found among scores between facilitators and experts for “inter-exchange experience with local and international cities” (40%) and “monitor and revise indicators” (37%) in mechanisms and processes, “evaluate performance of indicators and action plans” (37%) in outcome evaluations, and “interaction between government and community” (46%) and “interaction between civil organization and senior society” (39%) in resource integration. Clearly, facilitators showed overly optimistic assessments in AFC mechanisms and processes, outcome evaluation, and resource integration. The results showed disconnect between experts’ expectations versus actual practice conducted by facilitators. Implications of these findings are to integrate top down expectations with the realities of bottom up practice to design more realistic evaluations; continue to educate stakeholders about design, implementation and evaluation; and further integrate resources from government, civil organizations, and community.

## 1. Introduction 

Echoing the global trend of rapid elderly population growth, Taiwan’s 65-and-older population is expected to surpass 14% by 2018, officially making Taiwan an “aged society” [1]. Population aging is likely to produce a variety of economic, medical, family, and social impacts, including increasing healthcare burden associated with chronic illnesses and higher risk of disabilities. In an effort to respond to the potential impacts of population aging, in 2010 Taiwan’s government adopted the World Health Organization’s (WHO) age-friendly city (AFC) program strategies [2,3] for implementation by each of the country’s 22 local authorities. Based on previous scientific findings, a central goal of these efforts was to help older adults be physically and socially active and engaged in their communities, reflecting concepts of “active aging” and “age in place” [4,5]. Communities initiating age-friendly programs were expected to offer older residents a broad range of strategies, products, services and activities, including affordable housing, accessible transportation, intergenerational initiatives and opportunities, effective information and communication systems reaching community residents of all ages, safe and attractive outdoor spaces and public buildings, as well as social participation with family, friends and neighbors [6].

The AFC program built upon the success of Taiwan’s healthy city (HC) program, employing a health in-all-policies (HiAP) approach in policy-making. HiAP is an approach on health-related rights and obligations. It improves accountability of policymakers for health impacts at all levels of policy-making and includes an emphasis on the consequences of public policies on health systems, determinants of health, and well-being [7]. In 2010, Chiayi city became the first in the nation to adopt this program, and was followed by eight others the following year. The Dublin Declaration on Age Friendly Cities and Communities was initially developed in collaboration with the WHO in 2012 and mayors in 22 Taiwan cities co-signed the statement and made a commitment to the AFC program. Each local authority selected a program facilitator, who had the responsibility to implement or organize a local AFC program, sponsored by the central government. However, many local authorities experienced challenges or problems implementing an AFC program, including the following: no well-organized AFC leadership council, insufficient integration of resources between public and private sectors, lack of recognition for AFC program, indistinct mechanism or process to implement change, and inconsistent indicators or action plans based on need assessments of local elderly residents. Problems such as these reflected the fact that local program facilitators were not familiar with the AFC model, and lacked the capability or training to organize a new program such as this. 

To overcome these limitations, Taiwan’s Department of Health Promotion (DHP) sponsored a Taiwan Alliance for Healthy Cities (TAHC) to assist local authorities in implementing the AFC program. A team of expert consultants, consisting primarily of academic scholars and specialists in the AFC program, was charged with providing counseling and guidance to facilitators in all 22 local authorities regarding community health building. TAHC periodically organized meetings or seminars to exchange experiences and provided evidence needed to implement the AFC program [8]. Furthermore, Taiwan’s DHP sponsored an award competition, in which the expert team evaluated case studies from local authorities on performance or achievement in the eight AFC domains. Expert team members also provided training to facilitators in local authorities regarding implementation processes using the PDCA (plan, do, check and action) model, generating AFC indicators and action plans, searching and integrating local resources, and conducting needs assessment for seniors. The PDCA model is an iterative four-step management method used in organizations for the control and continuous improvement of processes and products. A logic model was also used for AFC program evaluation, which assumes that the way the program is designed or organized and how it is implemented affects the outcomes [9]. Information about the eight AFC domains was also compiled as teaching materials and placed on the TAHC website. Actual implementation of the AFC program seemed to be hampered by differing perspectives of the facilitators and the expert team. The evaluation done by the expert consultants allowed the facilitators to examine the implementation of the AFC program in order to provide continuous improvement of the program in the local government. This article examines the difference between the perspectives of the facilitators and expert consultants regarding the implementation AFC program, including the mechanisms and processes, outcome evaluation, and integration of resources. 

## 2. Materials and Method

### 2.1. Selection of Study Population 

Twenty-two facilitators from the local government in Taiwan were invited to participate in regular group meetings for the AFC program, sponsored by the HPA and TAHC. Over 80% of facilitators represented public health agencies from the local government, and the rest represented the administrative or planning agencies. The responsibility of facilitators included the following: organizing all program schedules, conducting need assessments for seniors, integrating and collaborating with other public and private sectors, encouraging community participation and stakeholder involvement, creating indicators and action plans, and conducting outcome evaluations. They played a key role as city project leaders or promoters for implementing the AFC program, and they determined whether or not the program was successful in the local governments. In addition, twenty inter-disciplinary scholars from the community health, public health, urban planning, gerontological nursing and architecture fields were invited for long-term collaboration with local authorities to counsel, guide, and assess the AFC program. TAHC and HPA organized expert team based on experiences of building community health and evaluating the quality of the program. Prior to implementing the AFC program, most of the experts had already established good partnerships with the local governments through various meetings and training programs, convening to periodically discuss the process of the AFC program. Meanwhile, the HPA and principal investigator (PI) of the AFC program usually held discussions with the experts to understand the progress and the level of performance in each local government. The well-organized commitment of the AFC program by the expert team was established incrementally through education and training using case studies, as well as during various meetings during the five-year period. The study was not submitted to the institutional review board due to low concern of harm to participants and the information being used to strengthen the implementation of the AFC program in the local government. All participants were still requested to sign the informed consent and anomalously completed the questionnaire.

### 2.2. Development of the Instrument 

Our instrument included three parts and was based on the WHO Ageing and Life Course Department, which provided the institutional leadership for the WHO Global AFC Network. It comprised of 20 items on the mechanism and process of the AFC program (including PDCA process), 23 items on outcome evaluations (including seven domains on political commitment, inter-sectorial collaboration and cooperation, community participation and stakeholder involvement, indicator and action plan, information and marketing, empowerment and training, and outcome evaluation) and 10 items on the integration of resources. These items were built on the initial work described by the implementation guidelines and from the minimum set of standards for communities that wish to participate in the WHO Global Age-Friendly Cities Network [2,3]. Each participant was invited to fill out the questionnaire anonymously after group meetings at different times and locations. The Likert scale was used to assess each item with five response categories: “strongly agree” represented 5 points; “agree” represented 4 points; “no opinion” represented 3 points; “disagree” represented 2 points and “strongly disagree” represented 1 point. The “very agree” and “agree” were pooled to be an “agreement to implement the activity”. Our study calculated the percentage of agreement in each item between facilitators and experts. Because the study used administrative data that were produced for periodic review by the central government, and all respondents participated in this survey anonymously, our proposal did not require review by an institutional review board. SPSS package 20 was used to calculate the percentage of agreement and average mean (SD) in each item in the three parts, and the calculations were used to compare the differences between facilitators and experts. The t-test was used to compare the average score of each item between facilitators and experts. Any difference was considered significant at *α* level of <0.05.

## 3. Results

Table 1 indicated there was considerably lower performance on the mechanisms and processes of the AFC program from the opinions of the experts, compared to facilitators, in the local authorities. Notably, 29% of great difference in the “plan stage” were found in “Organize education/training program” and “Establish commitment for officers”. In the “do stage”, slightly great differences on “Involvement and support from experts and specialist” and “Promote and initiate action plan by each unit” were 21% and 18%, respectively. Finally, the “check and action stage” had slightly great difference in the two groups, with “Assess impact and outcomes” (29%), “Participate in award competition” (26%) and “Upgrade efficiency in local authorities” (21%). 

Seven domains, based on the WHO guideline, had key indicators in implementing the AFC program. Findings were shown in Table 2. There were different distributions of the outcome evaluations on implementing the AFC program between the expert team and the facilitators, with four items with small difference (<10%), eight items with moderate difference (10%–20%), six items with slightly great difference (20%–30%) and five items with great difference (>30%). Generally, expert team scored considerably lower on each item in the outcome evaluation than the facilitators in the local government. Great difference (>30%) was found in five items, including “Inter-exchange experience with local and international cities” (40%), “Monitor and revise indicators” (37%), “Evaluate performance of indicators and action plans” (37%), “Recognized and supported by public opinion” (31%), and “Install age-friendly website” (30%). 

Integration of resources from the public and private sectors as well as stakeholder involvement in developing the age-friendly community are needed in the “plan stage” for effective implementation of the AFC program. Table 3 showed the integration of resource ratings by experts and facilitators. The items with significant difference between experts and facilitators included “Interaction between government and community” (46%), “Interaction between civil organization and senior society” (39%), “Interaction between inter-government units” (28%), “Interaction between government and non-governmental organization (NGO)/non-profit organization (NPO)” (29%) and “Interaction between task force and civil organization”. 

## 4. Discussion

This is the first article on elaborating the difference of implementing AFC program between two stakeholders. Three parts including mechanism and process, outcome evaluation, and integration of resources in Taiwan are completely consistent with the criteria from the WHO Global Network of Age-Friendly Cities and Communities program model and consists of other potential contributions to city age friendliness. In practice, different checklist approaches have been adopted or recommended [9]. Our framework model initiatorily applied a top-down approach on implementing the AFC program in local governments to better adapt their structures and services to the needs of their ageing population [10,11]. When the AFC program was launched in Taiwan, the council of steering committee was organized from the public sector, NGOs/NPOs, senior stakeholders, delegates from seniors and academia. Most of the political leaders supervised and guided the progress of the entire AFC program, based on reliable and valuable information monitored by the facilitators from the local authorities. However, our findings showed significant difference during the “plan and do” stages between the two groups, especially on “organize education/training program” (29%), “establish commitment for officers” (29%), and ”involvement and support from experts and specialist” (21%). It means that political leaders or facilitators in Taiwan may not entirely understand the conceptualization of AFC to order to make effective and plausible movement ahead. They need to not only focus on novel reform movements regarding civil services [12] but also effectively solve the realistic barriers and challenges for the elderly. Practically, implementing an AFC movement needs a well-organized platform to closely dialogue on inter-organizational plans and budget support by strong and cohesive leadership from political leaders. Similarly in Western countries, Menec et al [13] and Keating et al [14] also contend that what makes a successful AFC is having seamless congruence between the elderly’s need and the living environment, not conformity with a standard and fixed set of features. 

Instead of just an assessment of the facilitators from the local government, Taiwan has provided a multi-way assessment for implementing the AFC program. Need assessments of the life and health status for Taiwanese seniors from their perspectives has been periodically conducted by the central government and academia [4,15,16]. A national survey on the satisfaction of implementing the AFC program among Taiwanese seniors showed that 90% of seniors did not experience discrimination or exclusion other than by his family because of their age. Over 70% of seniors reported as very satisfied or satisfied in the seven domains of the AFC program. The highest satisfaction in the seven domains was “community and health services”, followed by “respect for the elderly and social integration”, but lowest were “work and volunteer service” and “barrier-free and safe public space” [17]. It is vital to understand the effective direction or strategy of implementing the AFC program by facilitators in the local governments, who play both provider and modifier roles. 

For the sake of fulfilling the vision of adopting “active ageing” and “age in place”, the AFC program in Taiwan should sustainably promote and broadly spread from the local government to various communities. The program should not only concentrate on the needs of the elderly population, but the program’s facilitators or promoters should also be periodically monitored by the central government on providing person-environment fit with high quality, accessible and available facilities in the community. Using a “strictly bottom-up” alternative approach in Taiwan may empower older persons to bargain for actual opportunities in cities to maintain their quality of life [18]. However, most of the elderly do not clearly perceive the AFC framework. Experts and those in academia are playing crucial roles regarding accompanying, instructing, and evaluating the operations of the AFC framework in local governments. There are mutual cooperation partnerships between facilitators and experts who are implementing the AFC program. Compared to the uneven, fragmentary implementation of the AFC program in Brazil [19], the AFC program in Taiwan has a well-organized framework put in place by the central government. The implementation of the program is ambitious, encompassing, and legally binding. If political leaders or facilitators in local governments do not conform to public policy from the central level on implementing the AFC program, they will not create a successful and sustainable age-friendly environment due to discouraged community participation and stakeholder involvement, inappropriate allocation of resources, and ineffective impact or outcome in communities. Unfortunately, previous studies failed to discuss the difficulty of implementing the AFC program from different stakeholders’ viewpoints [18,19]. Two critical factors for the successful and sustainable implementation of the AFC program are good leadership development and strong partnerships [5]. The strengths in the findings indicated that several domains and indicators including the mechanism and process, outcome evaluation, and integration of resources on implementing the AFC program. Based on the evaluating criteria, local governments should assess the current process and to then strengthen the leadership of political leaders or facilitators to encourage community participation and stakeholder involvement, and to fully allocate resources.

To date, political leaders in local governments have always had power to decide the design and progress of all policies and the allocation of budget and resources in Taiwan. However, our findings indicate a significant difference between the two stakeholders pertaining to “organize education/training program” (29%) and “establish commitment for officers” (29%) in the “planning stage.” Furthermore, there were also differences between the two stakeholders in “involve and support by experts and specialist” (21%) in the “do stage,” and “participate award competition” (26%), “assess impact and outcomes” (29%) and “upgrade efficiency in local authorities” (23%) in the “check and action stages.” These facts obviously indicated lack of time is well-prepared good commitment in teamwork and an accurate assessment of the outcomes and impacts of the local governments during the short-term period. If facilitators implement an AFC program that is not fully supported by leaders or does not negotiate with public or private sectors, then the promotion of the AFC program is unrealistic and at times redundant with other similar urban initiatives. Regrettably, political leaders in local governments are constantly changing in Taiwan, which can result in changes in policy interests and challenges or uncertainties to focusing on implementing the age-friendly plan [20]. As seen by Western countries, the importance of having leadership for successful AFC development is shown by the implementation evaluations in Canada [21] and the province of Manitoba [22]. No doubt strong political leadership and partnership are needed to support AFC initiatives [20].

The possible reasons for the great difference between two stakeholders are indicated as follows: (1). the different recognition of the AFC program existing; (2). most of facilitators had overly optimistic thinking in implementing the program; (3). the experts’ requirement of the facilitators meeting high standards and quality for the AFC program; (4). the need for facilitators need to do AFC work with multidisciplinary cooperation with other sectors; the work cannot be done independently; (5). because the outcome of the AFC program was difficult to be accurately evaluated by other sectors, experts were usually involved to guide the facilitators. We believe the experts provided objective assessments that can be used by local authorities; (6). outcomes of the AFC program are needed to be regularly evaluated by different stakeholders to improve quality, including older adults, delegates in the community, expert consultants, other institutions, or mass media, etc.; (7). The facilitators’ scores might be more subjective, because they might want a higher score in order to impress political leaders. However, the study found the less than 10% difference and more than 30% difference between the facilitators in local governments and scholars in the expert team, our findings can act as a reference for implementing the AFC program globally. It was based on the logic model, which is a suitable tool for program planning and evaluation purposes. Using a logic model like PDCA with the iterative process, facilitators in local governments may modify indicators in their logic models as often as necessary to implement the AFC program thoughtfully and sustainably. In the current study, items in Table 1, Table 2 and Table 3 are used as a checklist of the AFC program to evaluate the performance of planning and evaluation by facilitators in local governments or third parties [23]. Notably, five pillars requiring attention on implementing AFC program were identified: (1) the reinforcement of commitment for political leaders in local governments; (2) the empowerment of AFC skills and perception among facilitators; (3) the need to actively involve older persons in community planning and development actions; (4) the requirement of support and counseling from experts; (5) the effective allocation of resources, and the establishment of monitoring and assessing impact or outcomes of implementing AFC program [24]. Creating the vision and strategies of the AFC program are also required in creating a healthy city program [25].

## 5. Conclusions

Facilitators’ ratings of performance were consistently lower than experts’ ratings across all three domains of the questionnaire. Twenty-three out of 53 items showed a high level of difference, and the factors influencing the differences should be explored to determine how to minimize the discrepancies to enhance performance in the future. It is important to strengthen the leadership of political leaders, provide sufficient education/training program for facilitators, empower community participation and stakeholder involvement, and involve support from experts and specialists. Sustainable development of performance and accountability when implementing an AFC program is vital to fulfill the vision of “active ageing” and “ageing in place.” 

## Figures and Tables

**Table 1 ijerph-16-00608-t001:** Mechanisms and process ratings by experts and facilitators.

Criteria	Experts % (Mean ± SD)	Facilitators % (Mean ± SD)	Difference
Plan stage			
1. Well-operate task force	63% (3.53 ± 1.04)	61% (3.61 ± 0.77)	2%
2. Organize education/training program	32% (3.35 ± 0.84)	61% (3.57 ± 0.88)	29%*
3. Establish commitment for officers	32% (3.31 ± 1.1)	61% (3.41 ± 1.03)	29%*
4. Conduct need assessment for elderly	74% (3.89 ± 0.64)	69% (3.86 ± 0.76)	5%
5. Search and integrate local resources	58% (3.53 ± 0.6)	65% (3.61 ± 0.71)	7%
6. Encouragement and support from mayor or councilor	58% (3.58 ± 1.09)	70% (3.83 ± 0.64)	12%
7. Supervision by steer committee	53% (3.47 ± 0.99)	65% (3.70 ± 0.80)	13%
8. Set-up priority of elder problems	53% (3.32 ± 0.98)	52% (3.52 ± 0.73)	1%
9. Create and monitor indicators	58% (3.68 ± 1.03)	60% (3.61 ± 1.01)	2%
10. Generate action plan	42% (3.53 ± 1.04)	61% (3.65 ± 0.87)	19%
Do stage			
11. Assess the performance in each unit	37% (3.17 ± 1.01)	48% (3.35 ± 0.70)	11%
12. Make well-organized evaluation system	48% (3.50 ± 1.07)	52% (3.41 ± 0.89)	4%
13. Involvement and support from experts and specialist	53% (3.63 ± 0.93)	74% (4.00 ± 0.72)	21% *
14. Collaborate and cooperate with public and private sectors	42% (3.22 ± 0.92)	52% (3.22 ± 0.98)	10%
15. Promote and initiate action plan by each unit	48% (3.37 ± 0.87)	66% (3.39 ± 0.77)	18%
Check and Action stages			
16. Exchange inter-city empirical experience	38% (3.42 ± 1.09)	43% (3.59 ± 0.94)	5%
17. Participate in award competition	53% (3.63 ± 0.98)	79% (3.96 ± 0.75)	26% *
18. Encourage community participation	53% (3.53 ± 0.82)	56% (3.70 ± 0.80)	3%
19. Upgrade efficiency in local authorities	42% (3.26 ± 1.07)	65% (3.64 ± 0.57)	23% *
20. Assess impact and outcomes	37% (3.32 ± 0.98)	66% (3.74 ± 0.61)	29% *

* *p* < 0.05.

**Table 2 ijerph-16-00608-t002:** Outcome evaluation ratings by experts and facilitators.

Items	Experts % (Mean ± SD)	Facilitators % (Mean ± SD)	Difference
1. Political commitment			
A. Get commitment and support from mayor	58% (3.58 ± 1.09)	78% (3.96 ± 0.62)	20% *
B. Steering committee with inter-sectorial participation	58% (3.58 ± 0.94)	40% (3.78 ± 0.72)	12%
C. Recognized and supported by public opinion	21% (3.06 ± 0.80)	52% (3.43 ± 0.65)	31% **
2. Inter-sectoral collaboration and cooperation			
A. Operated and implemented by task force	68% (3.68 ± 0.92)	78% (3.91 ± 0.72)	10%
B. Setup vision, mission and strategy	69% (3.84 ± 0.81)	70% (3.78 ± 0.72)	1%
C. Generate regulation and code	32% (3.33 ± 0.82)	47% (3.43 ± 0.71)	15%
D. Establish network and platform	47% (3.47 ± 0.94)	70% (3.83 ± 0.87)	23% *
E. Periodical meeting with inter-sectorial unit	37% (3.21 ± 1.1)	61% (3.61 ± 0.92)	24% *
3. Community participation and stakeholder involvement			
A. Make effective mechanisms and awards	47% (3.37 ± 0.81)	52% (3.57 ± 0.71)	5%
B. Encourage involvement from seniors	52% (3.42 ± 0.82)	65% (3.59 ± 0.65)	13%
C. Periodical monitoring with need assessments from seniors	47% (3.47 ± 0.68)	61% (3.86 ± 0.76)	14%
D. Setup priority of action plan	37% (3.33 ± 1.11)	43% (3.36 ± 0.71)	6%
4. Indicators and action plan			
A. Provide training courses for facilitators	37% (3.53 ± 0.88)	56% (3.52 ± 0.97)	19%
B. Monitor and revise indicators	32% (3.58 ± 0.99)	69% (3.82 ± 0.89)	37% **
C. Generate action plan	37% (3.63 ± 0.98)	65% (3.82 ± 0.94)	28% *
D. Evaluate performance of indicators and action plans	11% (3.42 ± 1.09)	48% (3.50 ± 0.84)	37% **
5. Information and marketing			
A. Search and integrate various resources	27% (3.11 ± 0.87)	39% (3.18 ± 0.83)	12%
B. Install age-friendly website	11% (2.89 ± 1.1)	43% (3.27 ± 0.81)	32% **
6. Empowerment and training			
A. Empower knowledge and skill	32% (3.37 ± 0.93)	52% (3.50 ± 0.99)	20% *
B. Inter-exchange experience with local and international cities	21% (3.26 ± 0.85)	61% (3.55 ± 1.08)	40% **
7. Outcome assessment			
A. Collect input and process indicators	26% (3.42 ± 0.94)	52% (3.55 ± 0.66)	26% *
B. Monitor impact and outcome indicators	32% (3.16 ± 0.87)	47% (3.36 ± 0.83)	15%
C. Assess health status and quality of life for seniors	58% (3.53 ± 0.6)	52% (3.59 ± 0.98)	6%

* *p* < 0.05; ** *p* < 0.01.

**Table 3 ijerph-16-00608-t003:** Integration of resource ratings by experts and facilitators.

Items	Experts % (Mean ± SD)	Facilitators % (Mean ± SD)	Difference %
1. Interaction between experts and scholars	69% (3.84 ± 0.81)	83% (4.04 ± 0.62)	14%
2. Interaction between inter-government units	37% (3.42 ± 0.88)	65% (3.65 ± 0.63)	28% *
3. Interaction between government and community	37% (3.26 ± 0.64)	83% (3.87 ± 0.61)	46% **
4. Interaction between government and NGO/NPO	32% (3.29 ± 0.57)	61% (3.64 ± 0.64)	29% *
5. Integrate resources between inter-government	48% (3.28 ± 0.8)	52% (3.57 ± 0.71)	4%
6. Interaction between chief directors from government	37% (3.17 ± 0.76)	74% (3.83 ± 0.70)	37% **
7. Interaction with other cities	21% (2.89 ± 0.81)	30% (3.23 ± 0.79)	9%
8. Interaction between mayor and task force	42% (3.47 ± 0.88)	65% (3.78 ± 0.66)	13%
9. Interaction between task force and civil organization	32% (3.11 ± 0.74)	52% (3.59 ± 0.72)	20% *
10. Interaction between civil organization and senior society	31% (3.21 ± 0.77)	70% (3.82 ± 0.57)	39% **

* *p* < 0.05; ** *p* < 0.01.

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
