# Peer review of "Examining Stakeholder Perspectives: Process, Performance and Progress of the Age-Friendly Taiwan Program"

_ijerph, 2019, doi:10.3390/ijerph16040608_

Round 1
Reviewer 1 Report
Kia ora
If this program is about improving the holistic lives of older people – how were they involved in the inception, implementation and evaluation of this program? I am reminded of the mantra – “nothing about us, without us”.
As a health promotion specialist to me it is unclear how this program and evaluation engage with health promotion values such as equity, social justice, empowerment etc. If the program and evaluation are engaged with these fundamental values I suggest you make this explicit within your text. For me this engagement is part of what makes something a health promotion initiative.
This paper in its current form does not attempt to summarise the literature in the field or make explicit what this papers’ contribution will be and/ or what gap it is addressing.
The English academic writing is compromised throughout the paper making the paper difficult to read and understand. There is no clear overview of the program and what it is trying to achieve nor clear details of what the evaluation questions were or how it was conducted.
All the best with your future work.
Title - I suggest you change the title of the paper to be explicit what type of evaluation it is and remove reference to who conducted the evaluation.
Abstract – please avoid using acronyms in your abstract (and the text) of your paper. They reduce readability. I suggest you rework your abstract. Here are some suggestions of what to include that I always find helpful.
1. What is the driving question/issue/problem?
2. What do you mean by that? (Defining your terms/context)
3. For example….. (Quote, tangible, ‘show’ what you mean, concrete example of issue in the context of your work)
4. How so? (Methodological, relate to your work)
5. So what? (Implications, significance of paper/research, What will this paper add/contribute to the body of knowledge)
At the moment your abstract is not clear. Some of the key rational for your paper lies in the first paragraph of the text and should be summarised in the abstract which should be able to stand alone.
Introduction – suggest you make a clearer statement about what were the goals and objectives of the program and your evaluation.
P2ln63 – specialists in what?
P2n69 - where did the PDCA model come from? Suggest you include a reference.
P2Ln79 – It is unusual to start a sentence with a number – suggest rework.
Method and materials – some of the English academic writing in this section is problematic and weak and needs to be reworked. I have reread this section several times and I remain unclear about the detail, the nuts and bolts of your evaluation. Did the evaluation consist of a survey – if so was it a written, on-line, telephone survey – how many participants. It should be clear to me how you conducted your evaluation and why you did it that way.
Results – again some of the English academic writing in this section needs further work.
P3ln117 – I am still unclear who makes up your expert team.
I think most of your measures/indicators in all of the tables could be clearer.
Discussion – why is it important that the “expert” and the local facilitators had different perceptions of how the program is going? I suggest rewrite this section to focus on one idea per paragraph. At the moment it is very unclear and I believe unsuitable for publication.
Conclusion – this needs to be strengthened and aligned with the abstract.
Author Response
Dear Editors,
Thank you for your comments regarding our manuscript entitled “Evaluation of process on age-friendly Taiwan program by facilitators and experts”. We have responded to each of your comments on a case-by-case basis using bold words.
Reviewer 1
If this program is about improving the holistic lives of older people – how were they involved in the inception, implementation and evaluation of this program? I am reminded of the mantra – “nothing about us, without us”.
As a health promotion specialist to me it is unclear how this program and evaluation engage with health promotion values such as equity, social justice, empowerment etc. If the program and evaluation are engaged with these fundamental values I suggest you make this explicit within your text. For me this engagement is part of what makes something a health promotion initiative.
This paper in its current form does not attempt to summarize the literature in the field or make explicit what this papers’ contribution will be and/ or what gap it is addressing.
The English academic writing is compromised throughout the paper making the paper difficult to read and understand. There is no clear overview of the program and what it is trying to achieve nor clear details of what the evaluation questions were or how it was conducted.
All the best with your future work.
Response: Thank for your encouragement and suggestion. We will revise the text to meet your requirement as soon as possible.
Title - I suggest you change the title of the paper to be explicit what type of evaluation it is and remove reference to who conducted the evaluation.
Response: The title has changed to “Examining stakeholder perspectives: Process, performance and progress of the Age-friendly Taiwan Program”.
Abstract – please avoid using acronyms in your abstract (and the text) of your paper. They reduce readability. I suggest you rework your abstract. Here are some suggestions of what to include that I always find helpful.
1. What is the driving question/issue/problem?
2. What do you mean by that? (Defining your terms/context)
3. For example….. (Quote, tangible, ‘show’ what you mean, concrete example of issue in the context of your work)
4. How so? (Methodological, relate to your work)
5. So what? (Implications, significance of paper/research, What will this paper add/contribute to the body of knowledge)
Response: Thank for your comments, we have amended the abstract clearly. Please see the revised article.
At the moment your abstract is not clear. Some of the key rational for your paper lies in the first paragraph of the text and should be summarized in the abstract which should be able to stand alone.
Introduction – suggest you make a clearer statement about what were the goals and objectives of the program and your evaluation.
Response: Because AFC program in Taiwan was implemented by 22 local authorities through the federal government, skills or knowledge on AFC program for facilitators in local authorities are limited and met various obstacles from interdepartmental cooperation and community engagement. So, the federal government invited experts who had experience with building community health and evaluating the quality of the AFC program. The experts were trained by Taiwan Alliance for Healthy Cities (TAHC) and made the commitments to the AFC program to guide the facilitators in the local government. Our study is to try to evaluate the differences between the stakeholders (facilitators) and experts on the performance or outcome of the AFC program. These differences can help improve the quality of the AFC program in the future. Of course, another study evaluates the AFC satisfaction levels of older adults from a national survey that was launched by the federal government to monitor the outcomes or performance on the AFC program. Please see the revised article.
P2ln63 – specialists in what?
Response: Specialists were much more experienced in building community health and evaluating the quality of the program. Specialists are from public health, community health, gerontological nursing, urban planning, and architecture backgrounds. Please see the first paragraph of page 2 in the revised article.
P2n69 - where did the PDCA model come from? Suggest you include a reference.
Response: Thank for your suggestion, we have added the reference in the text. Please see the second paragraph of page 2 in the revised article.
P2Ln79 – It is unusual to start a sentence with a number – suggest rework.
Response: Thank for your suggestion, we have revised the text. Please see the last paragraph of page 2 in the revised article.
Method and materials – some of the English academic writing in this section is problematic and weak and needs to be reworked. I have reread this section several times and I remain unclear about the detail, the nuts and bolts of your evaluation. Did the evaluation consist of a survey – if so was it a written, on-line, telephone survey – how many participants. It should be clear to me how you conducted your evaluation and why you did it that way.
Response: Twenty-two local authorities implemented the AFC program and were counseled by 20 expert consultants. Each participant was invited to fill out the questionnaire anonymously after group meetings at different times and locations. Two experts were assigned to guide two local authorities and filled out the questionnaire anonymously. Please see the second paragraph of page 3 in the revised article.
Results – again some of the English academic writing in this section needs further work.
P3ln117 – I am still unclear who makes up your expert team.
Response: I am the principle investigator (PI) of AFC program sponsored by HPA from the central government. The expert team was selected and invited by the PI and HPA based on experiences of building community health and evaluating the quality of the program. Please see the first paragraph of page 3 in the revised article.
I think most of your measures/indicators in all of the tables could be clearer.
Discussion – why is it important that the “expert” and the local facilitators had different perceptions of how the program is going? I suggest rewrite this section to focus on one idea per paragraph. At the moment it is very unclear and I believe unsuitable for publication.
Response: Our study findings indicated the difference between facilitator and expert consultant evaluations on the performance and outcome of AFC program, and our findings can be used by the facilitators to improve the AFC program. We do believe the evaluations from expert consultants were objective judgments that helped improve the quality of AFC program in local government. We consistently found lower scores and percentages in each item evaluated by expert consultants. We hope these findings can help facilitators understand why experts gave lower scores for the program so they can improve the program.
Conclusion – this needs to be strengthened and aligned with the abstract.
Response: We will strengthen the findings and make it consistent with the abstract. Please see the conclusion in the revised article.
Reviewer 2 Report
Strengths
I commend the authors for conducting a study of the mechanisms and processes, outcomes, and integration for the age-friendly program in Taiwan. Exploring key learnings about process and evaluation are important for sustainability and innovation within and across communities and countries developing age-friendly programs and practices. There is a gap in the research literature on implementation and evaluation and particularly, with respect to exploring different stakeholder perspectives about performance and progress. This study makes an effort to address this gap.
Additional strengths of the manuscript include that it is well organized according to manuscript requirements, references are current and predominantly comprehensive, the purpose of comparing stakeholder perspectives is consistent throughout the manuscript, and the findings section consistently focuses on the three parts of the questionnaire and compares the stakeholder groups.
Comments for the authors consideration
Language
- there are extensive English language errors throughout the manuscript which detract from the value of the work
Title
- the title does not seem to substantially depict the full purpose of the study
- the following would be more comprehensive “Examining stakeholder perspectives: Process, performance and progress of the Age-friendly Taiwan Program”
Abstract
- it was not clear why the highlighted findings were reported (e.g., some were 30% or higher but others weren’t and their percentage was not reported)
- the last sentence could also refer to the need to revise indicators to better align with the local practice or context since this seems to be part of the explanation for the findings
Introduction
- Age-friendly work is grounded in a positive aging perspective rather than the burden of older adults on society. As such, it would be timely to insert a sentence at the beginning of line 4 that acknowledges the opportunities related to an aging population and contributions of older adults when provided with a supportive and inclusive environment.
Development of the Instrument
- the authors mention that the 53 items questionnaire was based on the WHO ALCD but it was unclear which components were included/excluded and why or whether new items were added
- providing an example of a question for each part of the questionnaire would assist the reader with interpreting the findings
- further details about the data collection process (e.g., questionnaire distribution) are needed
- it is mentioned that “for the sake of comparison” the 2 agreement ratings were combined but this did not seem to be relevant since the mean was provided, as such this sentence could be removed
- lines 112 – 114 seemed out of place and would fit better in the results section
Results
- the analysis process was not described (e.g., why did you decide to calculate the mean and standard deviation, how was significance determined, why did you not use t-tests, did sample size influence the statistics you selected)
- how did you decide on cut-off points for small vs. great differences (e.g., less than 10%, 30% or higher) and were there other studies that also used these or did you create them
- the write up for each part of the questionnaire was inconsistent in format which made it difficult to determine which findings were important and how this was determined
- the discussion related to Table 2 seemed the most balanced by providing a summary of little to great differences and particularly drawing attention to those findings with 20% or greater differences
- summarizing or noting differences that were less than 10% (low) and greater than 20% and 30% would be useful information for each part of the questionnaire/section of the results since a comparison is about what is both different and similar
- include what the items were assessing in the title of each table (e.g., Table 1: Mechanisms and processes: Comparison of stakeholder perspectives)
Discussion
- I found that the discussion section highlighted some results but not necessarily the main findings (e.g., paragraph 1)
- Paragraph 2 didn’t seem relevant except to reiterate what was also said in paragraphs 3 & 4 about bottom up vs. top down policy and practice. As such, the paragraph didn’t seem relevant.
- Diverse explanations for the main findings could have been presented.
- Below are some observations I noted from the results that I expected to see in the discussion:
1) The facilitators had higher ratings on each questionnaire item (except 1) than the experts
- Why was this consistently the case?
- Could this be due to the facilitators feeling pressure to rate themselves at a high level to maintain their jobs (e.g., performance reviews could result in inflated ratings)?
- Could this be due to their belief that they were in fact meeting expectations based on the training they received and the work they had completed?
- Did experts believe they needed to be stricter with their assessments (e.g., as part of their role as experts)?
2) 23 out of 53 items showed a high level of difference
- This is a lot of items with a high level of difference which suggests that something fundamental is not aligned, especially since this is across 22 local authorities.
- Could this be due to a community process being used by the facilitators but indicators representing policy rather than a practice reality (paragraphs 3 & 4 refer to this)?
- Is it possible that the experts may need to spend more time in the field to more accurately determine what is suitable performance and progress in actual practice?
- Is it a problem if the two stakeholder groups have different perspectives? Was the original hypothesis that they would be similar? Perhaps experts will always judge more strictly?
3) 5 items from part B and 3 items from part C had differences that were 30% or higher
- Are there themes across these items? What do they collectively tell us?
- Are these differences we would expect to see or do they contradict strengths of the program?
- Are good leadership and strong partnerships the answer to addressing these differences? How might this reduce the differences between facilitators and experts? Is there another explanation?
4) 14 of the 53 items had differences of less than 10% (which suggest a shared perspective)
- What can be learned from these findings?
- How might this information be useful for progress in other areas?
- What could be recommendations arising from these findings?
5) Which items had the lowest % (e.g., and which had the highest %?
- What can be learned from these findings?
- What could be recommendations arising from these findings?
Conclusion
- Line 234: the performance wasn’t lower, just the perception of performance by the experts was lower
- I would suggest revising the conclusion to highlight the main results based on less than 10% difference (e.g., set up, assess needs, plan/do/act, integrate resources, interact with other cities) and more than 30% difference (e.g., 4 integration items, awareness and support from the public, revise indicators) and what these findings mean for local and international age-friendly program facilitators and experts who are engaged in this process.
References
- Include Age-friendly Cities: A Guide (WHO, 2007) since this is a foundational document.
Author Response
Dear Editors,
Thank you for your comments regarding our manuscript entitled “Evaluation of process on age-friendly Taiwan program by facilitators and experts”. We have responded to each of your comments on a case-by-case basis using bold words.
Reviewer 2
Strengths
I commend the authors for conducting a study of the mechanisms and processes, outcomes, and integration for the age-friendly program in Taiwan. Exploring key learnings about process and evaluation are important for sustainability and innovation within and across communities and countries developing age-friendly programs and practices. There is a gap in the research literature on implementation and evaluation and particularly, with respect to exploring different stakeholder perspectives about performance and progress. This study makes an effort to address this gap.
Additional strengths of the manuscript include that it is well organized according to manuscript requirements, references are current and predominantly comprehensive, the purpose of comparing stakeholder perspectives is consistent throughout the manuscript, and the findings section consistently focuses on the three parts of the questionnaire and compares the stakeholder groups.
Response: Thank for your encouragement and suggestions for the text. We have improved the content of the text.
Comments for the authors consideration
Language
- there are extensive English language errors throughout the manuscript which detract from the value of the work
Response: Thank for your suggestions, we have improved the English language errors in the content of the text.
Title
- the title does not seem to substantially depict the full purpose of the study
- the following would be more comprehensive “Examining stakeholder perspectives: Process, performance and progress of the Age-friendly Taiwan Program”
Response: The title has been changed to “Examining stakeholder perspectives: Process, performance and progress of the Age-friendly Taiwan Program”. Please see the title in the revised article.
- it was not clear why the highlighted findings were reported (e.g., some were 30% or higher but others weren’t and their percentage was not reported)
- the last sentence could also refer to the need to revise indicators to better align with the local practice or context since this seems to be part of the explanation for the findings
Response: We tried to revise the abstract and highlight the findings. Specifically, the items of great difference (30% or higher) between facilitator and expert consultant should be examined. We encourage the facilitator to find the cause of difference and the reasons for the lower expert consultant scores and percentages to improve AFC program in the future. We have also provided a checklist for facilitators to successfully implement the AFC program. Please see the abstract in the revised article.
Introduction
- Age-friendly work is grounded in a positive aging perspective rather than the burden of older adults on society. As such, it would be timely to insert a sentence at the beginning of line 4 that acknowledges the opportunities related to an aging population and contributions of older adults when provided with a supportive and inclusive environment.
Response: Thank you for your comment to explain the reasons of implementing AFC program. The significance the AFC program was grounded in the increase of the aging population and the promotion of healthy aging. We tried to modify the first paragraph in the P1. Please see the first paragraph of page 1 in the revised article.
Development of the Instrument
- the authors mention that the 53 items questionnaire was based on the WHO ALCD but it was unclear which components were included/excluded and why or whether new items were added
- providing an example of a question for each part of the questionnaire would assist the reader with interpreting the findings
Response: From items in Table 1-3, the reader can easily understand the mechanism and process as well as the outcome assessment and resource integration for AFC program. A logic model was also used for AFC program evaluation, which assumes that the way the program is designed or organized and how it is implemented affects the outcomes. Each step is evaluated by several items and also by inter-sectorial collaboration between experts and facilitators. Please see the second paragraph of page 2 in the revised article.
. - further details about the data collection process (e.g., questionnaire distribution) are needed
- it is mentioned that “for the sake of comparison” the 2 agreement ratings were combined but this did not seem to be relevant since the mean was provided, as such this sentence could be removed
- lines 112 – 114 seemed out of place and would fit better in the results section
Response: We have amended the sentence and made more clarification in the text.
Results
- the analysis process was not described (e.g., why did you decide to calculate the mean and standard deviation, how was significance determined, why did you not use t-tests, did sample size influence the statistics you selected)
Response: We have added the statistical method in the revised text. Please see the second paragraph of page 3 in the revised article.
- how did you decide on cut-off points for small vs. great differences (e.g., less than 10%, 30% or higher) and were there other studies that also used these or did you create them
Response: Difference of 30% or higher for each item was labeled as “great difference” in the two groups. We highlighted the great difference in each item for the facilitator to find the reason of difference in order to improve the quality of the AFC program. Although difference of less than 10% had no significant difference for both groups, items with low scores still need to be recognized to improve the AFC program. For example, even though the “Set-up priority of elder problems” had a 1% difference between both groups, the scores only reflected an “average score” (scores were 3.32 and 3.52 out of 5). For this item, only 53% of experts and 52% of facilitators agreed that this item was done well. This shows that there needs to be more improvement to encourage older adults to participate in the AFC program. Please see the second and third paragraph of page 7 in the revised article.
- the write up for each part of the questionnaire was inconsistent in format which made it difficult to determine which findings were important and how this was determined
Response: Questionnaire included 53 items in three parts, with 20 items for the mechanism and process of the AFC program, 23 items for outcome evaluation, and 10 items for integration of resources (Please see Table 1 -3). The three parts were used to evaluate the structure, performance, and outcome for the AFC program. If the total score in each part was high, it represented a good performance of implementing the AFC program. In the study, 53 items in three parts was used for implementing AFC in other cities and the items that scored the lowest were needed to improve the implementation of the program. The central government needs to provide support to change the obstacles in implementing the AFC program in the local government. Please see the second paragraph of page 3 in the revised article.
- the discussion related to Table 2 seemed the most balanced by providing a summary of little to great differences and particularly drawing attention to those findings with 20% or greater differences
Response: Yes, Table 2 showed significant differences of 20% or greater in the two groups. These items should be given more attention to find the factors in improving the implementation of the AFC program. We added the items with 20% or greater significant differences and explained the reason for the differences in the discussion. Please see the third paragraph of page 7 in the revised article.
- summarizing or noting differences that were less than 10% (low) and greater than 20% and 30% would be useful information for each part of the questionnaire/section of the results since a comparison is about what is both different and similar
Response: We highlighted the items with differences greater than 20% and 30% in order to identify useful reasons of difference to improve the quality of the program. Items with differences less than 10% still had their average scores checked to see if the scores were low or high to see if those items needed to be improved. Experts had lower scores than facilitators because the facilitators were overly optimistic or overestimated the program. The experts still considered that each item needed to be improved. Please see the third paragraph of page 7 in the revised article.
- include what the items were assessing in the title of each table (e.g., Table 1: Mechanisms and processes: Comparison of stakeholder perspectives)
Response: Thank you for your suggestion. We have amended the title.
Discussion
- I found that the discussion section highlighted some results but not necessarily the main findings (e.g., paragraph 1)
Response: Paragraph 1 emphasizes our findings and their significance. Previous studies failed to discuss the difficulty of implementing the AFC program from both the experts’ and facilitators’ viewpoints. Please see the first paragraph of page 7 in the revised article.
- Paragraph 2 didn’t seem relevant except to reiterate what was also said in paragraphs 3 & 4 about bottom up vs. top down policy and practice. As such, the paragraph didn’t seem relevant.
- Diverse explanations for the main findings could have been presented.
- Below are some observations I noted from the results that I expected to see in the discussion:
Response: We revised the paragraphs in the discussion section.
1) The facilitators had higher ratings on each questionnaire item (except 1) than the experts
- Why was this consistently the case?
- Could this be due to the facilitators feeling pressure to rate themselves at a high level to maintain their jobs (e.g., performance reviews could result in inflated ratings)?
- Could this be due to their belief that they were in fact meeting expectations based on the training they received and the work they had completed?
- Did experts believe they needed to be stricter with their assessments (e.g., as part of their role as experts)?
Response: The discussion section added the reasons why facilitators had higher ratings for each item. Experts usually used a higher standard when evaluating the AFC program, emphasizing high quality in each step to achieve successful outcomes. Please see the last paragraph of page 7 in the revised article.
2) 23 out of 53 items showed a high level of difference
- This is a lot of items with a high level of difference which suggests that something fundamental is not aligned, especially since this is across 22 local authorities.
- Could this be due to a community process being used by the facilitators but indicators representing policy rather than a practice reality (paragraphs 3 & 4 refer to this)?
- Is it possible that the experts may need to spend more time in the field to more accurately determine what is suitable performance and progress in actual practice?
- Is it a problem if the two stakeholder groups have different perspectives? Was the original hypothesis that they would be similar? Perhaps experts will always judge more strictly?
Response: The findings not only showed 23 out of 53 items with a high level of difference, but we also elaborated on the low scores and percentages for each item from both stakeholders. Facilitators also were encouraged to find the reasons for low scores and factors that affected them to improve the quality of implementing the AFC program. Please see the first paragraph of page 8 in the revised article.
3) 5 items from part B and 3 items from part C had differences that were 30% or higher
- Are there themes across these items? What do they collectively tell us?
- Are these differences we would expect to see or do they contradict strengths of the program?
- Are good leadership and strong partnerships the answer to addressing these differences? How might this reduce the differences between facilitators and experts? Is there another explanation?
Response: The great differences for each item were provided by the study. The factors that should be addressed by the facilitators in the local government were identified to discuss the next steps in implementing the AFC program. The possible reasons for the great differences were summarized as follows: 1. The different perceptions of the AFC program that existed between the two stakeholders, 2. Most of the facilitators had overly optimistic thinking in the implementation of the program, 3. The experts required the facilitators to meet high standards for the AFC program, 4. The facilitators need to be more effective in multidisciplinary cooperation with other sectors, because the work cannot be done independently, 5. Because it was difficult to accurately evaluate the outcome of the AFC program by other sectors, the experts were usually involved in guiding the facilitators in the local government. We believe the experts provided objective assessment in evaluating the program, which can be used by the local government, 6. Outcomes of AFC program need to be regularly evaluated by different stakeholders including older adults, delegates in community, expert consultants, other institution or mass media, etc. to improve the quality of the program, 7. The facilitators’ scores might be more subjective, because they might want a higher score in order to impress political leaders. Please see the third paragraph of page 7 in the revised article.
4) 14 of the 53 items had differences of less than 10% (which suggest a shared perspective)
- What can be learned from these findings?
- How might this information be useful for progress in other areas?
- What could be recommendations arising from these findings?
Response: Although items with 10% difference or lower did not have significant difference in the study, the items with lower scores or percentages still need to be highlighted to improve the program. For example, items 5 or 7 in Table 3. The lower scores or percentages for each item among facilitators or experts should be highlighted and improved in the future.
5) Which items had the lowest % (e.g., and which had the highest %?
- What can be learned from these findings?
- What could be recommendations arising from these findings?
Response: The explanation is similar to question 4. The message from the findings is to take action to improve the quality for AFC program in the local government. Please see the first paragraph of page 8 in the revised article.
Conclusion
- Line 234: the performance wasn’t lower, just the perception of performance by the experts was lower
- I would suggest revising the conclusion to highlight the main results based on less than 10% difference (e.g., set up, assess needs, plan/do/act, integrate resources, interact with other cities) and more than 30% difference (e.g., 4 integration items, awareness and support from the public, revise indicators) and what these findings mean for local and international age-friendly program facilitators and experts who are engaged in this process.
Response: You are right. The conclusion needs to highlight the items with great difference and low scores or percentages and to improve the quality of the AFC program. Please see the first paragraph of page 8 in the revised article.
References
- Include Age-friendly Cities: A Guide (WHO, 2007) since this is a foundational document.
Reviewer 3 Report
This is a very interesting paper about an internationally important topic. Some clarifications and tightening of the writing (along with extensive copy editing) will help this paper make a solid contribution to the literature.
The introduction is well-written and contains appropriate and useful background. However, some of the concepts and terms can benefit from a bit more detail, assuming that some international readers will not have in-depth familiarity with them. Specifically, could you add a sentence to explain: “Health-in-all-policies” approach, and the Dublin Declaration?
In the methods section, it isn’t clear how the 22 facilitators in local authorities were recruited for the group meeting, and whether 22 is a large or small percentage of possible participants, and whether they were a good representation of the pool of possible participants. A bit more detail about the “20 scholars from expert team” would also be helpful; what is the role and composition of the expert team?
My major concern is that it was never quite clear what exactly was asked of the respondents. The PDCA stages and domains are described, but what exactly were the participants assessing? Were they as about whether this action should be taken, about whether this action was taken, and/or about how important this action is to the success of AFC? There is a sentence in the results section suggesting that the respondents were assessing performance of the TAHC on these items, but then there is mention of “perceptions on implementing”, which could mean perceptions about what should happen or about what did happen. The result of this lack of clarity is that it is difficult to understand what it means to say that one group “scored higher”. Writing of results and discussion could also be clearer.
Finally, in the discussion section, it will help the reader if you can clarify the links among views of elders, a grass-roots approach to AFC, and the findings of this study. The issue of how such efforts should be initiated and implemented is very important, and the complexities of leadership, authority, and grass-roots stakeholders (elders themselves) is introduced nicely in this section, but it just needs to be clearer.
Author Response
Dear Editors,
Thank you for your comments regarding our manuscript entitled “Evaluation of process on age-friendly Taiwan program by facilitators and experts”. We have responded to each of your comments on a case-by-case basis using bold words.
Reviewer 3
This is a very interesting paper about an internationally important topic. Some clarifications and tightening of the writing (along with extensive copy editing) will help this paper make a solid contribution to the literature.
Response: Thank you for your encouragement. We should provide the information in implementing the AFC program.
The introduction is well-written and contains appropriate and useful background. However, some of the concepts and terms can benefit from a bit more detail, assuming that some international readers will not have in-depth familiarity with them. Specifically, could you add a sentence to explain: “Health-in-all-policies” approach, and the Dublin Declaration?
Response: Thank you for your suggestions. We have added the description of HiAP and Dublin Declaration in the introduction section. Please see the second paragraph of page 2 in the revised article.
In the methods section, it isn’t clear how the 22 facilitators in local authorities were recruited for the group meeting, and whether 22 is a large or small percentage of possible participants, and whether they were a good representation of the pool of possible participants. A bit more detail about the “20 scholars from expert team” would also be helpful; what is the role and composition of the expert team?
Response: In the revised text, we have added the background of the 20 scholars from the expert team. Please see the revised text. Please see the first paragraph of page 3 in the revised article.
My major concern is that it was never quite clear what exactly was asked of the respondents. The PDCA stages and domains are described, but what exactly were the participants assessing? Were they as about whether this action should be taken, about whether this action was taken, and/or about how important this action is to the success of AFC? There is a sentence in the results section suggesting that the respondents were assessing performance of the TAHC on these items, but then there is mention of “perceptions on implementing”, which could mean perceptions about what should happen or about what did happen. The result of this lack of clarity is that it is difficult to understand what it means to say that one group “scored higher”. Writing of results and discussion could also be clearer.
Response: Yes, the implementation of the AFC program was not evaluated on the “perceptions on implementing.” We tried to emphasis using evidence-based evaluations to achieve a high quality program. In order to have a higher score for each item, local facilitators needed to provide various reference materials and documents to support the responses. The revised text has been added in the Method section.
Finally, in the discussion section, it will help the reader if you can clarify the links among views of elders, a grass-roots approach to AFC, and the findings of this study. The issue of how such efforts should be initiated and implemented is very important, and the complexities of leadership, authority, and grass-roots stakeholders (elders themselves) is introduced nicely in this section, but it just needs to be clearer.
Response: Yes, a grass-roots approach to evaluating the outcome of AFC is needed, and we have done a survey by the older adult population in Taiwan to meet this need. In paragraphs 2 and 3 of the Discussion section, we mentioned the multiway assessment in implementing the AFC program, which was published in other journals. In Taiwan, there have been regular evaluations done by different stakeholders including older adults, delegates in community, expert consultants, other institution or mass media, etc. to improve the quality of the program. The findings have increased the awareness of political leaders and the older adult population. Please see the third paragraph of page 7 in the revised article.
Round 2
Reviewer 1 Report
Thanks for the changes you have made. This paper is stronger for the work you have done.
P2 Ln66 It is still not clear whether their is consumer eg. old people involvement in this project.
P2 Ln81 what makes a expert consultant? Please explain or moderate your claims of expertise. I think you have overused the term expert and might want to choose different language.
p2 ln83 this is getting clearer but why from an evaluation perspective are the key voices the expert consultants and the facilitators - what about the participants/ the older people?
p3 ln86 did this study get ethics approval from a recognised ethics committee? If not can you be clearer why not?
p3 ln252 Good to see you finally mentioning the value of older people contributing to this process. I am unsure why this isn't a stronger theme of this paper.
All the best with your future work.
Author Response
Dear Reviewer 1,
Thank you for your comments regarding our manuscript entitled “Evaluation of process on age-friendly Taiwan program by facilitators and experts”. We have responded to each of your comments on a case-by-case basis using bold words.
Reviewer 1
Thanks for the changes you have made. This paper is stronger for the work you have done.
Response: Thank you for your encouragement.
P2 Ln66 It is still not clear whether their is consumer eg. old people involvement in this project.
Response: Our team did not invite the older adults to participate due to involving expert consultants who guided the aging friendly city program for the facilitators from the local government. Please see the revised article.
P2 Ln81 what makes a expert consultant? Please explain or moderate your claims of expertise. I think you have overused the term expert and might want to choose different language.
Response: Thank you for your suggestion. The expert consultant was invited by the principle investigator and the central government. The expert played the role of consultant to guide and evaluate the performance of the facilitators from the local government. We kept using the term “expert consultant” to avoid confusion if other terms are used. Please see the revised article.
p2 ln83 this is getting clearer but why from an evaluation perspective are the key voices the expert consultants and the facilitators - what about the participants/ the older people?
Response: The study was to assess the facilitator’s performance or outcomes of implementing the AFC program by the expert consultant, which did not include other older adults. The focus of this study emphasizes the evaluation of the program by the experts and facilitators, not the older adults. Please see the revised article.
p3 ln86 did this study get ethics approval from a recognised ethics committee? If not can you be clearer why not?
Response: Our study is not reviewed by IRB. In the revised article we added our reasons. Please see at P3 Ln 109-113 in the revised article.
p3 ln252 Good to see you finally mentioning the value of older people contributing to this process. I am unsure why this isn't a stronger theme of this paper.
Reponses: This study emphasizes the evaluation of the impact of the AFC program by stakeholders. A manuscript in another Journal (Global Health Promotion) has submitted findings examining the perspectives of AFC program from the older population. Please see the revised article.
All the best with your future work.
Reviewer 2 Report
I appreciate the efforts of the authors to respond to my comments and suggestions, and provide a revised version. I have a few further comments for consideration.
Line 27-28: Could be replaced with "The results showed a disconnect between experts' expectations versus actual practice conducted by facilitators."
Line 28-29: This statement doesn't capture the various conclusions mentioned in the discussion. How about something like this: Implications of these findings are to integrate top down expectations with the realities of bottom up practice to design more realistic evaluations; continue to educate stakeholders about design, implementation and evaluation; and further integrate government, civil organizations, and community.
Line 52-55: Dublin Declaration is repeated three times
Line 120 & 122: very agree and very disagree should be replaced with strong agree and strongly disagree
Change Table titles for consistency to:
Table 1: Mechanisms and process ratings by experts and facilitators
Table 2: Outcome evaluation ratings by experts and facilitators
Table 3: Integration of resources ratings by experts and facilitators
Line 147-149: Remove non significant results for consistency with the reporting of the other tables.
Line 157-160: These two sentences are not suitable for inclusion in the results and should be moved to the discussion. This paragraph should be written like the other results paragraphs for consistency.
Line 222-224: This is not a complete sentence. This needs to be edited.
Line 244-256: I think these are very important points that highlight questions/comments I raised in my initial review. Is there a way to integrate these throughout the discussion rather than as a list?
Line 271-272: I still think this sentence needs to be rewritten. Perhaps: Facilitators' ratings of performance were consistently lower than experts' ratings across all three components of the questionnaire.
Line 272-275: Delete this sentence.
Line 277: replace "elaborated on to improve the quality of each item in the future" with "explored to determine how to minimize the discrepancies to enhance performance in the future."
Line 280-281: I suggest revising this sentence and replacing it with something specific about performance and accountability when implementing an AF program.
Thank you for sharing your findings about this program! This will help inform other countries working on evaluating their age-friendly programs.
Author Response
Dear Reviewer 2,
Thank you for your comments regarding our manuscript entitled “Evaluation of process on age-friendly Taiwan program by facilitators and experts”. We have responded to each of your comments on a case-by-case basis using bold words.
Reviewer 2
I appreciate the efforts of the authors to respond to my comments and suggestions, and provide a revised version. I have a few further comments for consideration.
Response: Thank you for your encouragement.
Line 27-28: Could be replaced with "The results showed a disconnect between experts' expectations versus actual practice conducted by facilitators."
Response: Thank you for your suggestions, and we have changed Line 27-28. Please see at P1 Ln 27-31 in the revised article.
Line 28-29: This statement doesn't capture the various conclusions mentioned in the discussion. How about something like this: Implications of these findings are to integrate top down expectations with the realities of bottom up practice to design more realistic evaluations; continue to educate stakeholders about design, implementation and evaluation; and further integrate government, civil organizations, and community.
Response: Thank you for your suggestions, and we agree with your suggestion to change Line 28-29. Please see at P1 Ln 18-29 in the revised article.
Line 52-55: Dublin Declaration is repeated three times
Response: Thank you for your suggestions, and we have changed Line 52-55. Please see the at P2 Ln 54-57 in revised article.
Line 120 & 122: very agree and very disagree should be replaced with strong agree and strongly disagree
Response: Thank you for your suggestions, and we have changed Line 120-122. Please see at P3 Ln 127-128 in the revised article.
Change Table titles for consistency to:
Table 1: Mechanisms and process ratings by experts and facilitators
Table 2: Outcome evaluation ratings by experts and facilitators
Table 3: Integration of resources ratings by experts and facilitators
Response: Thank you for your suggestions, and we have revised the Table titles. Please see Table in the revised article.
Line 147-149: Remove non significant results for consistency with the reporting of the other tables.
Response: Thank you for your suggestions, and we have revised Line 147-149. Because another reviewer suggested for us to indicate the items with small difference (<10%) and moderate difference (10%- 20%), we added them to highlight the discrepancies between the experts’ and facilitators’ opinions in the text. Please see at P4 Ln 151-153 in the revised article.
Line 157-160: These two sentences are not suitable for inclusion in the results and should be moved to the discussion. This paragraph should be written like the other results paragraphs for consistency.
Response: Thank you for your suggestions, and we have revised Line 157-160. The description is still needed to highlight the significance of Table 3. Please see at P5 Ln 161-164 in the revised article.
Line 222-224: This is not a complete sentence. This needs to be edited.
Response: Thank you for your suggestions, and we have edited Line 222-224. Please see at P7 Ln 227-230 in the revised article.
Line 244-256: I think these are very important points that highlight questions/comments I raised in my initial review. Is there a way to integrate these throughout the discussion rather than as a list?
Response: Thank you for your suggestions. We will keep our format to make it easier to clearly understand the difference between experts and facilitators. The factors should be used to improve the implementation of the AFC program and to try to minimize the discrepancies among stakeholders to enhance performance of the AFC program. Please see the revised article.
Line 271-272: I still think this sentence needs to be rewritten. Perhaps: Facilitators' ratings of performance were consistently lower than experts' ratings across all three components of the questionnaire.
Response: Thank you for your suggestions, and we have revised Line 271-272. Please see at P8 Ln 278-279 in the revised article.
Line 272-275: Delete this sentence.
Response: Thank you for your suggestions, and we have deleted Line 272-275. Please see at P8Ln 280-281 in revised article.
Line 277: replace "elaborated on to improve the quality of each item in the future" with "explored to determine how to minimize the discrepancies to enhance performance in the future."
Response: Thank you for your suggestions, and we have revised Line 277. Please see at P8 Ln 280-281 in the revised article.
Line 280-281: I suggest revising this sentence and replacing it with something specific about performance and accountability when implementing an AF program.
Response: Thank you for your suggestions, and we have revised Line 280-281. Please see at P8 Ln 284-285 in the revised article.
Thank you for sharing your findings about this program! This will help inform other countries working on evaluating their age-friendly programs.
Response: Your wonderful comments and suggestions helped improve our manuscript. We sincerely appreciated your efforts and contribution.